# Therapeutic Potential of *Lappula patula* Extracts on Germline Development and DNA Damage Responses in *C. elegans*

**DOI:** 10.3390/ph18010089

**Published:** 2025-01-13

**Authors:** Qinghao Meng, Anna Hu, Weiyu Xiao, Robert P. Borris, Hyun-Min Kim

**Affiliations:** 1Division of Natural and Applied Sciences, Duke Kunshan University, Kunshan 215316, China; 2School of Pharmaceutical Science and Technology, Tianjin University, Tianjin 300072, China

**Keywords:** *Lappula patula*, DNA repair, germline development, medicinal plants, herbs, meiosis

## Abstract

Background: *Lappula patula (L. patula*) is a plant with known medicinal properties, and its extracts have shown promise as potential anti-cancer agents. This study aimed to evaluate the nematocidal effects of L. patula extracts and investigate their impact on germline development, DNA damage responses, and apoptosis in *Caenorhabditis elegans (C. elegans*), a model organism for studying these processes. Methods: *C. elegans* was exposed to *L. patula* extracts to assess survival, development, and incidence of male phenotype. Germline abnormalities were examined using microscopy at different developmental stages. The DNA damage response was evaluated through the expression of the *atm-1, atl-1* and *pCHK-1*. Apoptosis was quantified by monitoring cell death during the pachytene stage. LC-MS was used to identify bioactive compounds within the extracts. Results: Exposure to *L. patula* extracts resulted in a dose-dependent reduction in worm survival and larval developmental progress, with no significant impact on the male incidence. Germline defects were observed, including increased nuclear spacing at premeiotic and pachytene stages, altered number of bivalents during diakinesis. These defects correlated with a significant decrease in brood size. Also, *L. patula* extracts activated the DNA damage response pathway, marked by increased expression of *atm-1* and *atl-1*. Moreover, the extracts induced apoptosis in the germline in a pCHK-1-independent manner. LC-MS analysis revealed 31 potential anti-tumor compounds, supporting the extract’s cytotoxic properties. Conclusions: *Lappula patula* extracts exhibit potent nematocidal and cytotoxic properties, suggesting their potential for cancer therapy. The observed DNA damage and apoptosis in *C. elegans* emphasize the extract’s promising role in anti-cancer drug development. Further studies are needed to explore the therapeutic potential of these compounds in clinical settings.

## 1. Introduction

Previously, we conducted a screening of herbal extracts to assess their potential anti-tumor effects using *C. elegans* [1]. Of the extracts tested, 16% showed reduced survival rates and induced larval arrest or lethality, suggesting that larval arrest plays a crucial role in determining worm viability. Notably, a small subset of herbal extracts, including those from *Lappula patula*, *Onobrychis cornuta*, and *Torenia species*, exhibited a remarkable increase in male occurrence and reduced survivability. Building on these observations, this report focuses on *Lappula patula* to investigate its effects on growth, germline development, DNA damage checkpoints, and repair mechanisms in the *C. elegans* model.

### 1.1. General Overview of the Lappula Genus (Boraginaceae)

The genus *Lappula*, belonging to the family Boraginaceae, consists of approximately 50–70 species [2,3] and is predominantly distributed across Eurasia, North Africa, the Americas, and Australia [4,5]. These species thrive in temperate regions, often inhabiting dry, sandy soils, grasslands, hillsides, and disturbed areas. For instance, *L. squarrosa*, a rare species, is sporadically found in Central Europe. Central Asia, particularly Xinjiang Province in China, is recognized as a diversity hotspot for *Lappula* species [4,5]. *Lappula patula* in particular is recognized as a species with a fragmented and localized distribution within certain regions. In some areas, it has been noted as a floristic rarity and regional novelty, with occurrences in limited or peripheral environments [6].

*Lappula* plants typically feature blue or white flowers with five throat appendages, a subulate gynobase, and nutlets arranged in clusters of four. These nutlets, often equipped with marginal wings or glochids, facilitate dispersal by attaching to animals or surfaces [7]. *L. patula* differs from its closely related species, *L. botschantzevii*, in both the size of the corolla and the structure of the inflorescence [8].

### 1.2. Taxonomy and Phylogeny

The taxonomy of *Lappula* has undergone significant refinement over time. Early classifications identified 38 species based on nutlet morphology [9]. Subsequently, Popov et al. classified 39 species in the Flora USSR and introduced a more detailed infrageneric system [7]. In the 21st century, Ovczinnikova et al. expanded the classification to 70 species [3,4]. However, molecular phylogenetic studies revealed that *Lappula* is polyphyletic, leading to the reassignment of some species to other genera, such as *Rochelia* and *Pseudolappula* [4,10]. In line with this, recent research highlights the genus’s considerable diversity, particularly in northwestern China [7].

### 1.3. Medicinal Properties and Uses

Several *Lappula* species are known for their medicinal properties. For instance, *Lappula myosotis* is used to treat wounds and joint issues due to its anti-inflammatory and insecticidal effects [11]. Similarly, *Lappula echinata*, prevalent in northern China, exhibits antioxidant and immunomodulatory effects, including protection against oxidative damage in macrophages [12]. Its extracts also demonstrate antibacterial activity, with a quinolone alkaloid found to be effective against pathogens like *Pseudomonas pyocyanea* and *Staphylococcus epidermidis* [11]. Traditional uses of *L. echinata* include treating chronic diarrhea, with studies indicating its ability to inhibit intestinal smooth muscle contractions [13].

Additionally, *Lappula* species contain polyunsaturated fatty acids, such as stearidonic acid (SDA), which may provide anti-inflammatory and cardiovascular benefits [14]. However, the presence of pyrrolizidine alkaloids (PAs) in *Lappula* poses safety risks due to their genotoxic and carcinogenic properties [14].

Despite these risks, the medicinal potential of the *Lappula* genus remains promising. *C. elegans* is an excellent model system for assessing the toxicity of herbal compounds. Its simplicity, coupled with its genetic similarity to humans, conserved biological pathways, and advanced genetic tools makes it invaluable for toxicity studies [15,16,17,18].

### 1.4. Findings on Lappula patula

In this study, we observed that *Lappula patula* extracts activated a DNA damage checkpoint response via ATM and ATR pathways in a pCHK-1 independent manner. This response disrupted germline development and induced pachytene apoptosis, indicating impaired DNA damage repair mechanisms. Further analysis revealed a reduction in bivalents and disrupted meiotic progression, highlighting the adverse effects of *Lappula patula* extracts on developmental processes.

LC-MS analysis identified 112 compounds in the *Lappula patula* extracts, 31 of which possess known anti-tumor activity. Notably, linoleic acid was the only compound overlapping with *Onobrychis cornuta*, a plant previously identified for its potent anti-tumor properties. The abundance of anti-tumor compounds in *Lappula patula* underscores its broader therapeutic potential.

## 2. Results

### 2.1. Nematocidal Effects of Lappula patula Extracts

The extracts from *Lappula patula* (*L.p*) demonstrated a significant reduction in the survival rates of *C. elegans* compared with the untreated group, with a decrease comparable to that observed with *Onobrychis cornuta* (*O.c*), a plant previously studied in our laboratory (Figure 1A: 89 versus 38 for +DMSO and *O.c*., respectively; 89 versus 40 for +DMSO and *L.p*. at 0.03 µg/mL, respectively). This reduction in survival suggests that *Lappula patula* contains active compounds which impair the health and viability of the nematodes. Furthermore, the exposure to *Lappula patula* extracts also resulted in larval arrest or lethality, similar to what was induced by the *O. cornuta* extracts (Figure 1A: 91 versus 45 for +DMSO and *O.c*., respectively; 91 versus 42 for +DMSO and *L.p*. at 0.03 µg/mL, respectively). These findings suggest that *Lappula patula* exhibits potent nematocidal activity, which may be linked to growth defects caused by the bioactive compounds within the extract.

In *C. elegans*, errors in the segregation of sex chromosomes during meiosis can lead to offspring with an irregular sex chromosome composition. This abnormal segregation often leads to an increased occurrence of males, a condition referred to as high incidence of males (HIM). The HIM phenotype serves as a well-established marker for identifying disruptions in chromosome segregation and meiotic progression, making it an important tool for studying the impacts of various environmental or chemical factors in reproductive health and genetic stability [16,19]. In our study, exposure to *Lappula patula* extracts resulted in a significant elevation in the HIM phenotype, indicating that the extract may interfere with the normal process of meiotic division and disrupt the proper segregation of sex chromosomes in worms. Specifically, the worms treated with *Lappula patula* extracts showed an increase in the number of males within the population, similar to the results observed with *O. cornuta* extracts (Figure 1A, 0.3 versus 4 for +DMSO and *O.c*., respectively; 0.3 versus 4 for +DMSO and *L.p*. at 0.03 µg/mL of *L.p*. extracts, respectively). This suggests that *Lappula patula* may be exerting its toxic effects by influencing chromosome behavior, possibly through the induction of DNA damage or the alteration of key proteins involved in meiotic processes.

In addition to the observed nematocidal effects, the increased HIM phenotype implies that *Lappula patula* could be causing underlying genetic instability in the worms. The disruption of normal meiotic processes, which can lead to chromosome mis-segregation, is a well-known hallmark of genomic instability and can result in the development of various diseases, including cancer. The fact that *Lappula patula* extracts induce such phenotypic changes raises the possibility that this herb might contain compounds which not only affect the viability of nematodes but could also be relevant for studying the mechanisms of DNA damage and chromosomal abnormalities in more complex organisms.

These results highlight the potential of *Lappula patula* as a valuable source of bioactive compounds, which could be instrumental in investigating genetic and meiotic integrity. We explored further to investigate and identify the specific compounds responsible for these effects and explore the broader implications of *Lappula patula* extracts.

### 2.2. Dose-Dependent Nematocidal and Larval Arrest of Lappula patula Extracts

To further explore and confirm the nematocidal effects of *Lappula patula* (*L.p*.) extracts, we conducted a series of experiments to examine whether different concentrations of the extract were correlated with observed changes in *C. elegans* phenotypes. This approach was inspired by previous studies on *Onobrychis cornuta* (*O.c*.) extract, which also exhibited dose-dependent effects on worm survival and larval development [1]. As the concentration of the herbal extracts increased, we observed a progressive decline in survivability, representing a clear dose-dependent relationship (Figure 1A). For instance, in the case of *O. cornuta*, the survival rate decreased from 38% to 30% and finally to 22%, with concentrations of 0.03, 0.3, and 3 µg/mL, respectively. Similarly, for *Lappula patula* extracts, the survival rate dropped from 40% to 22% and then to 20% with increasing concentrations of 0.03, 0.3, and 3 µg/mL, respectively. These results demonstrate a strong correlation between extract concentration and worm survival, supporting the hypothesis that higher doses of *Lappula patula* extracts significantly impact nematode viability.

In addition to the survival rate, the number of adult worms also declined as the concentration of the herb extracts increased, suggesting that the extracts disrupt mitotic growth and hinder proper development. Specifically, in *Lappula patula*-exposed worms, the percentage of adults dropped from 42% to 36% and then to 30% as the concentration increased from 0.03 µg/mL to 0.3 µg/mL and 3 µg/mL, respectively. Similarly, in *Onobrychis cornuta*-treated worms, the percentage of adults decreased from 45% to 41% and ultimately to 32% while following the same concentration gradient. This decline in adult worm numbers further reinforces the idea that *Lappula patula* extracts interfere with normal mitotic division and cellular proliferation in *C. elegans*.

Although we observed the induction of the high incidence of males (HIM) phenotype—characterized by an increase in the proportion of male worms—following exposure to both *Onobrychis cornuta* and *Lappula patula* extracts, we did not detect a clear dose-dependent pattern for this phenotype. This suggests that induction of the HIM phenotype may not be directly tied to the concentrations of the herbal extracts, and other factors may contribute to the observed changes in sex chromosome segregation (Figure 1A). In our study, the HIM phenotype in *Onobrychis cornuta* was observed at frequencies of 4%, 5.3%, and 3.7% for the 0.03, 0.3, and 3 µg/mL concentrations, respectively, while in *Lappula patula*, the HIM incidence was found to be 4%, 4.7%, and 2.2% for the same concentrations. This indicates that although both extracts induced the HIM phenotype, the extent of induction was not directly proportional to the dose of the extract.

Taken together, our findings highlight a clear dose-dependent relationship between the *Lappula patula* extract concentration and the survival rate and larval arrest. However, the induction of the HIM phenotype did not follow a dose-dependent pattern, suggesting that other mechanisms may be at play in influencing sex chromosome segregation in response to herbal exposure. These observations emphasize the potential nematocidal and cytotoxic nature of *Lappula patula*.

### 2.3. Impact of Lappula patula Extracts on Nematode Survival and Bacterial Growth

One potential explanation for the observed decrease in *C. elegans* survival upon exposure to *Lappula patula* (*L.p*.) extracts could be a secondary effect related to bacterial growth, which serves as the primary food source for *C. elegans* in laboratory settings. Given that *C. elegans* relies on *E. coli* OP50 for nutrition, it is essential to confirm that the observed nematocidal effects are not confounded by bacterial inhibition. To test this hypothesis, we conducted experiments to assess whether *Lappula patula* extracts exert any inhibitory effects on the growth of *Escherichia coli* OP50, the bacterial strain used in *C. elegans* culturing.

We incubated *E. coli* OP50 with *Lappula patula* extracts for 24 h and measured bacterial growth by monitoring the optical density (OD) at 600 nm. The results revealed no significant difference in bacterial proliferation between the *Lappula patula*-treated group and the DMSO control group at both 12 and 24 h of incubation. Specifically, at the 12 h time point, the optical density was 0.32 for the OP50 + DMSO group and 0.25 for the OP50 + *Lappula patula* extract group (Figure 1B, *p* = 0.2432), indicating no substantial inhibition of bacterial growth. At the 24 h mark, the OD values were 0.46 for the OP50 + DMSO group and 0.41 for the OP50 + *Lappula patula* extract group (*p* = 0.2854), further reinforcing the observation that the herbal extract did not significantly impact bacterial growth under the conditions tested. This finding suggests that the reduced survival of *C. elegans* upon exposure to *Lappula patula* extracts is not due to a defect in bacterial growth but rather reflects a direct impact of the extract on nematode health.

Taken together, these findings support the hypothesis that *Lappula patula* extracts significantly impair *C. elegans* survival and development, including larval arrest and compromised sex chromosome segregation. The observed dose-dependent decrease in worm survivability and the induction of larval arrest further strengthen the idea that *Lappula patula* extracts exert a potent nematocidal effect. This suggests that the extract targets essential biological processes in *C. elegans*, rather than affecting their food source. The correlation between an increased extract concentration and diminished survival, along with the observation of arrested larval development, highlights the potential of *Lappula patula* as a promising herbal candidate for further investigation into nematocidal properties, independent of its effects on bacterial growth.

### 2.4. L.p Extracts Cause Defective Germline Progression

In *C. elegans*, germline development is a highly regulated process, with nuclei undergoing a precise spatial and temporal arrangement as they advance through different stages of meiosis and mitosis. During normal germline progression, mitotic nuclei are located at the distal tip of the premeiotic zone (PMT) and gradually transition to the meiotic prophase as they move toward the transition zone (TZ), where nuclei begin to adopt a crescent shape. This shift marks the commencement of the meiotic process [16,20]. Given that exposure to *Lappula patula* extracts resulted in defective meiotic progression in our initial observations, we aimed to further investigate how these herbal extracts affect the overall development and organization of the germline in *C. elegans*.

To explore this, we dissected adult hermaphrodites, stained their gonads with DAPI, and examined the arrangement of the nuclei in the germline. In control animals, the nuclei were neatly organized and spaced during progression from the premeiotic tip to the pachytene stage. However, in *L. patula*-treated worms, we observed significant disruptions in the spacing between the nuclei in the PMT, TZ, and pachytene stages, suggesting that the extracts interfere with normal germline progression. These observations were confirmed by quantifying the distances between adjacent nuclei, which showed a significant increase in the gaps between the nuclei in both the PMT and pachytene stages (Figure 2A,B). Specifically, the distance between nuclei in the PMT stage was 4.3 µm in the controls, while it increased to 8.7 µm in the *L.p*-treated worms (*p* < 0.0001). Similarly, in the pachytene, the gap between the nuclei increased from 5.2 µm in the control group to 11.4 µm in the treated group (*p* < 0.0001). These data strongly suggest that *L.p* extracts cause significant disorganization and failure in the normal progression of germline nuclei. *O.c* extracts have also been reported to induce a similar phenotype.

The transition from mitosis to meiosis is marked by the appearance of crescent-shaped nuclei [16]. *L. patula* extracts altered the frequency of crescent-shaped nuclei in both the PMT and pachytene stages. In the PMT stage, the incidence of crescent-shaped nuclei was significantly increased in the *L. patula*-treated group compared with the controls (0.7% versus 3.1%, respectively; Figure 2A,C). Similarly, during the pachytene stage, the frequency of crescent-shaped nuclei was 1.2% in the control group and 3.7% in the treated group, indicating that the transition from mitosis to meiosis was impaired throughout the progression from the TZ stage to the pachytene stage. Also, *O.c* extracts have been reported to induce a similar phenotype in PMT.

While we observed similar disruptions in germline development with *Onobrychis cornuta* (*O.c*.) extracts, which led to the formation of chromatin bridges between gut cells (a hallmark of defective chromosomal segregation during anaphase or cytokinesis) [21,22], no such chromatin bridges were observed in the *L. patula*-treated worms. This indicates that the disruption caused by *L.p.* extracts may not primarily manifest as mitotic defects in gut cells or other mitotic tissues but rather specifically impact germline development (Figure 2A).

As *C. elegans* enters the diakinesis stage of meiosis, six bivalents formed by homologous chromosomes, held together by chiasmata, become visible under DAPI staining meiosis [16]. However, when exposed to *L. patula* extracts, we observed a failure to form the typical six pairs of homologous chromosomes, with five or seven DAPI-stained bodies observed, indicating defective chromosomal segregation and DNA repair (Figure 2A,D). Specifically, in the control group, 7% of worms exhibited five DAPI-stained bodies, 93% displayed the expected six, and none showed seven. In contrast, the *L.p.*-treated group had 10% with five DAPI-stained bivalents, 86% with the expected six, and 7% with seven, which were absent in the control. These findings point to a clear defect in DNA repair mechanisms and chromosomal integrity, stemming from the *L. patula* extract treatment.

In addition to these observations of chromosomal and nuclear defects, we did not note a reduction in the overall size of the germline. The normal spatial organization of germline nuclei in *C. elegans* is indicative of healthy development, and defects in this organization often correlate with a reduction in germline size [21,23]. While exposure to *O.c* extracts resulted in a significant shortening of the PMT and pachytene stages compared with the control (Figure 2E, 59 µm versus 44 µm in PMT; 281 µm versus 213 µm in pachytene, *p* = 0.0021 and *p* = 0.0004, respectively), no significant changes in germline length were observed in the *L. patula*-exposed worms. This suggests that while *L. patula* extracts impair germline progression, they do not affect the overall length of the developmental stages, unlike *O. cornuta*.

Given the reduction in survival observed in the *C. elegans* exposed to *L. patula* extracts, we hypothesized that these defects in germline progression might also lead to a decrease in fertility. To test this, we quantified the brood size of hermaphrodites exposed to *L. patula* extracts over a period of four days starting from the L4 stage. While the control worms showed only a mild reduction in brood size on day 1, the *L. patula*-treated worms exhibited a significant drop in fertility starting on day 2, with brood sizes decreasing from 147 to 35 offspring per worm (Figure 2F, *p* = 0.0022). This reduction in fertility further supports the idea that *L. patula* extracts interfere with meiotic development, resulting in compromised reproductive success.

These cumulative observations demonstrate that *L. patula* extracts disrupt the orderly progression of the germline, starting with defects in meiotic progression and culminating in reduced fertility. The presence of crescent-shaped nuclei during the PMT and pachytene stages, along with the failure to form six pairs of homologous chromosomes during diakinesis, indicates defective DNA repair and improper chromosomal segregation. These abnormalities lead to a significant decrease in brood size, highlighting the detrimental impact of *L. patula* extracts on germline development and fertility. Collectively, our findings suggest that *L. patula* extracts disrupt normal germline progression and mitotic growth, ultimately impairing fertility in *C. elegans.*

### 2.5. L. patula Extracts Activate the DNA Damage Checkpoint Pathway, Resulting in Increased Apoptosis, Independent of pCHK-1 Involvement

The DNA damage response constitutes a critical signaling network which orchestrates cellular reactions to DNA lesions, initiating processes such as DNA repair, apoptosis, and cell cycle arrest [24,25,26]. Both ATM and ATR can activate CHK1 directly or via intermediary kinases, with CHK1 activation driving downstream responses which facilitate DNA repair, arrest cell cycle progression, and preserve genome integrity in response to DNA damage or replication stress.

We examined the levels of *atm-1* expression (a mammalian ATM hjomolog), *atl-1* (a mammalian ATR homolog), and pCHK-1 (the active form of CHK1, a mammalian CHK1 homolog). Treatment with *Lappula patula* resulted in increased expression of the major DNA damage checkpoint components, indicating that the herbal treatment activated the DNA damage response. Specifically, *atm-1* expression was induced 1.7 fold, while *atl-1* expression showed a 1.8 fold increase (Figure 3A, *p* = 0.0018 and *p* = 0.0006, respectively).

Interestingly, the downstream target of ATM and ATR, pCHK-1, did not show increased levels in the pachytene stage of the germlines (Figure 3B, 1.6 versus 1.6 for control and *L. patula*-treated samples, respectively, *p* = 0.825). In contrast, treatment with *O.c*. significantly induced pCHK-1 foci formation (1.6 versus 4.1 for control and *O.c*.-treated samples, respectively), suggesting distinct mechanisms of DNA damage checkpoint activation between the two herbal extracts [25,27].

Unresolved DNA intermediates can lead to apoptosis in pachytene-stage nuclei [27]. Consequently, we conducted further analysis of DNA damage-induced apoptosis in the germline. Compared with the untreated control group, which exhibited rather few acridine orange-stained nuclei, treatment with *Lappula patula* significantly increased the number of such nuclei during the pachytene stage (Figure 3C, 0.3 versus 1.2 for control and *L. patula*, respectively, *p* = 0.0008).

In summary, our findings demonstrate that exposure to *Lappula patula* extracts elevated the expression of key components within the ATM- and ATR-dependent DNA damage checkpoint pathways, leading to an increase in apoptosis. This indicates the activation of the DNA damage response. These results strongly suggest that persistent, unresolved DNA damage activates the checkpoint pathways, ultimately resulting in enhanced apoptosis within the germline of *C. elegans*. However, the absence of changes in the pCHK-1 foci levels upon *L. patula* treatment suggests that apoptosis induction occurred independently of pCHK-1, unlike *O. cornuta.*

### 2.6. LC-MS Analysis of Lappula patula Extract Identified 112 Bioactive Compounds, Including 31Anti-Tumor Constituents

Herb extracts contain many active compounds, including polyphenols, alkaloids, terpenoids, essential oils, tannins, and saponins. To identify the specific compounds involved in the DNA damage pathway, we performed liquid chromatography–mass spectrometry (LC-MS) analysis to isolate and characterize the active constituents of *Lappula patula* extracts.

LC-MS analysis of the *L. patula* extract identified a total of 112 compounds (Appendix A), which can be categorized into nine major groups (Table 1), each of which may contribute to the bioactive properties of the herb. The flavonoid category in the *L.p.* extract includes a diverse array of polyphenolic compounds such as rutin, quercetin, and luteoloside, which are renowned for their antioxidant, anti-inflammatory, and antimicrobial activities. These compounds contribute to cardiovascular protection, immune modulation, and cellular health. The presence of flavonoids like astragalin and tiliroside further highlights the extract’s potential for combating oxidative stress and inflammation. Terpenoids, represented by compounds such as ursolic acid, and β-caryophyllene exhibit a variety of bioactivities, including anti-inflammatory, anticancer, and antimicrobial properties. These compounds play vital roles in plant defense mechanisms and hold promise for therapeutic applications in human health. The extract also contains proanthocyanidins (tannins), including procyanidin A1 and procyanidin B1, which are known for their strong antioxidant properties and cardiovascular benefits. These compounds help mitigate oxidative stress and contribute to cellular protection. The inclusion of fatty acids and lipids, such as linoleic acid and palmitic acid ethyl ester, suggests roles in energy storage, skin health, and anti-inflammatory responses. These essential components underline the extract’s potential for supporting metabolic and physiological processes. Alkaloids, including talatisamine and ikarisoside D, are nitrogen-containing compounds with notable pharmacological effects, such as neuroprotection, pain relief, and antimicrobial activity. These compounds add to the therapeutic diversity of the *L.p.* extract. The presence of stilbenoids like trans-THSG and cis-THSG points to significant antioxidant and anti-aging properties, which contribute to cellular health and longevity. These phenolic compounds are crucial for maintaining metabolic balance and mitigating oxidative damage. Lignans, such as epipinoresinol and isopinoresinol, are plant-derived compounds with antioxidant and hormone-regulating activities. Their potential in cancer prevention and hormone modulation further emphasizes the bioactive nature of the extract. In the category of steroids and glycosides, compounds like ergosterol peroxide and oleuropein provide anti-inflammatory and antioxidant effects, enhancing the extract’s role in immune support and cellular defense. Finally, the miscellaneous compounds group, which includes betaine and uridine, highlights the extract’s ability to support metabolism, stress adaptation, and cellular energy production. Together, these diverse categories illustrate the multifunctional bioactive potential of the *Lp.* extract, offering promising therapeutic and protective benefits.

Notably, the *L.p* extract is particularly enriched with 31 anti-tumor active compounds, underscoring its potential in cancer prevention and treatment (Figure 4 and Table 2). These compounds, through their diverse bioactivities, suggest a synergistic effect which may enhance the extract’s efficacy in targeting cancer cells while supporting overall health. This remarkable profile positions the *L.p* extract as a valuable resource for further research and development in pharmaceutical and nutraceutical applications.

## 3. Discussion

This research utilized *C. elegans* to evaluate the nematocidal toxicity of *Lappula patula* herbal extracts and explore their effects on DNA damage repair and checkpoint responses for the first time (Figure 5). These findings suggest *L. patula* has anti-cancer potential, a topic not studied before, positioning it as a promising candidate for cancer therapeutics, warranting further research. A screening of 316 herbal extracts revealed *Lappula patula* as a potent inducer of DNA damage checkpoint activation, DNA damage mediated apoptosis, HIM phenotypes, impaired meiotic progression, and decreased survival rates [1,22].

The induction of male phenotypes appears to occur only within a certain concentration range, beyond which further increases in the dosage do not affect the incidence of males. This suggests that male phenotype induction is not a direct consequence of higher doses but may instead result from a saturation effect beyond a specific threshold. In contrast, the dose-dependent larval arrest observed indicates that the herb extract’s effects on worm survivability correlate with its concentration. As the dose increases, larval growth is inhibited, leading to reduced survival rates. This supports the idea that the dose of *L. patula* extract directly impacts larval development and survival, although it does not appear to further influence male incidence at higher doses.

While it is surprising that the *L. patula* extract induced pCHK1-independent apoptosis, unlike *O. cornuta* extracts, several studies have demonstrated CHK1-independent DNA damage-induced apoptosis. For example, inhibition of Chk1 in human tumor cells triggers hyperactivation of ATM, ATR, and caspase-2, leading to apoptosis after DNA damage [28]. This is supported by our findings, which show elevated levels of both ATM and ATR upon exposure to *L. patula*. Given the presence of numerous anti-tumor compounds in *L. patula*, the synergistic effects of the 31 identified anti-tumor compounds may contribute to the complexity of the mechanism. Therefore, we plan to investigate these mechanisms in more detail through systematic approaches in future studies.

### 3.1. Potential Anti-Tumor Properties of Lappula patula

The components identified in the *Lappula patula* extracts exhibited a variety of potential anti-tumor effects through different biological mechanisms. Flavonoids such as quercetin and rutin, along with phenolic compounds like ferulic acid, are well known for their strong antioxidant and anti-inflammatory activities [29]. These properties play a crucial role in tumor suppression and protection against oxidative stress, which can contribute to cancer prevention. Additionally, ursolic acid and oleanolic acid, two triterpenoids, have been shown to inhibit cell proliferation and metastasis, potentially playing a significant role in suppressing tumor growth and spreading [30,31].

The anticancer compounds identified in *Lappula patula* extracts play a role in the phenotypes observed in *C. elegans*. For example, linoleic acid, an essential omega-6 polyunsaturated fatty acid present in both *Lappula patula* and *Onobrychis cornuta*, mimicked the phenotypes induced by the herb extracts. These include increased apoptosis and activation of the MAPK signaling pathway. [1]. However, while the linoleic acid or *O.c* extracts induced the pCHK-1 foci level, the *L.p* extracts did not, also suggesting the difference between and complex mechanisms of the *L.p* and *O.c*.

This complexity may stem from the diverse compound composition of herbs. Meanwhile, most of the compounds identified in the *Torenia* species extracts were also present in the extracts of *Onobrychis cornuta* and *Veratrum lobelianum* [1,22]. However, only linoleic acid was found to be a common anti-tumor compound found among *L.p*, *O.c.*, and *V.I.*, suggesting a potential different mode in *L.p*. This may explain why *L.p.* exhibited pCHK-1 independent DNA damage-induced apoptosis, whereas the other three extracts did not.

### 3.2. Research and Application Potential

The diversity of the compounds identified in the *Lappula patula* extracts presents significant opportunities for future research and therapeutic applications. Notably, bioactive substances like ergosterol peroxide and limonin have the potential to serve as novel candidates for anti-cancer drug development, offering new avenues for therapeutic intervention [32,33]. These compounds may share similar mechanisms of action with other promising bioactive molecules by targeting key cellular processes involved in cancer progression. Moreover, when used in combination with existing anti-cancer therapies, they may produce synergistic effects, enhancing the overall efficacy of treatment. The broad chemical profile of these compounds opens up exciting possibilities for the development of innovative drugs and combination therapies which could improve cancer treatment outcomes while minimizing side effects.

The findings indicate that *L. patula* extract contains a wide variety of bioactive compounds, which may contribute to its therapeutic potential. These results provide a valuable foundation for further pharmacological research and offer insights into the diverse mechanisms through which *L. patula* may exert its health benefits. Moreover, to fully validate the specificity of *L. patula’s* effects and deepen our understanding of its molecular mechanisms, it will be essential to conduct comparisons with additional control compounds targeting similar pathways. This approach will help clarify whether the observed effects are specific to *L. patula* or if they are shared by other compounds with similar biological activities. Investigating these effects using other plant extracts or synthetic compounds which target the same pathways will not only confirm the distinctiveness of *L. patula’s* actions but also aid in identifying the specific pathways involved, which could be crucial for its potential application. Such studies would be critical in validating *L. patula* as a promising candidate for future therapeutic development and ensuring that its bioactive compounds are not only effective but also selectively modulate cancer-related pathways.

While this study highlights the therapeutic potential of *L. patella*, which contains 31 identified anti-tumor compounds, it is important to acknowledge the potential risks associated with its use. Specifically, the extract exhibits cytotoxic effects, including genotoxicity, which could lead to unintended consequences. These risks should be carefully considered, and further studies are needed to fully evaluate the safety profile of *L. patella* for therapeutic applications.

## 4. Materials and Methods

### 4.1. Strains and Alleles

All *C. elegans* strains were maintained at 20 °C as previously described [34]. The N2 obtained from the *Caenorhabditis* Genetics Center was used as a wild-type reference.

### 4.2. Herb Extraction

*Lappula patula* was collected from Armenia in May 2006 as described in [1,22]. In brief, the plant material was cleaned, air-dried, and ground into a coarse powder before being extracted with methanol. The methanol extract was concentrated, dissolved in 90% aqueous methanol, and extracted with n-hexane. The remaining hydroalcoholic phase was freed of methanol and sequentially extracted with dichloromethane and n-butanol to yield hexane, butanol, and water-soluble fractions (-H, -B, and -A, respectively). The final extracts in hexane were dissolved in DMSO and adjusted to a concentration of 1 mg/mL before being diluted in M9 buffer to a final concentration of 0.03 µg/mL for use in the experiments. The same sample was also applied to LC-MS (performed by Yanbo Times in Beijing, China). Each English name in the LC-MS results was translated based on the Chinese name provided by Yanbo Times and may not be entirely accurate. Please also consider other factors, including mass, extraction mass, and chemical formula, for a more comprehensive analysis.

### 4.3. Larval Arrest or Lethality, Survival, and HIM

Hermaphrodites were harvested from NGM plates to establish synchronized L1 stage larvae, following the protocols outlined in [25,35]. The larvae were then incubated in 180 µL of the herb extract solution and transferred to a 96 well plate. After gentle agitation, the plates were incubated at 20 °C for 24 h, with continuous monitoring of phenotypic changes for up to 48 h. To assess survival, worm mobility was tracked after 24 h of exposure to the herbal extracts. The brood size was defined as the total number of eggs laid by each individual worm over a 4–5 day period after the L4 stage. Larval arrest or lethality was determined as the percentage of hatched worms which did not survive to adulthood. The male proportion in the population was also calculated, representing the percentage of adult males in each group. Statistical significance between different genotypes was assessed using a two-tailed Mann–Whitney test (95% confidence interval). All experiments were repeated in triplicate. This protocol was adapted from Kim and Colaiacovo [35].

### 4.4. LC–MS Analysis

LC–MS analysis was performed as described previously [1,22]. The results were validated against a standard sample database. All compounds identified were confirmed using this rigorous method.

### 4.5. Monitoring the Growth of E. coli

The growth of *E. coli* OP50 in the presence of herb extracts was evaluated by measuring the optical density (OD) at 600 nm as previously described in [1,36]. The herb extracts were tested for their antibacterial effects by monitoring bacterial growth under 0.03 µg/mL of each extract.

### 4.6. Immunofluorescence Staining

Immunofluorescence staining of whole-mount gonads was performed as described in [25,27,37]. The primary antibodies used included rabbit anti-pCHK-1 (1:250, Cell Signaling, Danvers, MA, USA, Ser345), and the secondary antibodies were Cy3 anti-rabbit (1:300, Jackson Immunochemicals, West Grove, PA, USA).

### 4.7. pCHK-1 Foci

Measurement of the pCHK-1 foci was performed as described in [25,27]. From 5 to 10 germlines per treatment were analyzed. Statistical comparisons were conducted using a two-tailed Mann–Whitney test or a *t*-test, with a 95% confidence interval.

### 4.8. Quantitation of Germline Apoptosis

Germline apoptosis was assessed with acridine orange staining in age-matched (20 h post-L4) animals, following the method described in [38]. A Nikon Ti2-E fluorescence microscope was used to score between 20 and 30 gonads per treatment. Statistical significance was determined using the two-tailed Mann–Whitney test with a 95% confidence interval.

### 4.9. Quantitative Real-Time PCR

The cDNA was synthesized from RNA extracted from young hermaphrodite worms using an ABscript II First Strand synthesis kit (ABclonal, Woburn, MA, USA, RK20400). Real-time qPCR was performed using ABclonal 2X SYBR Green Fast Mix (RK21200) in a LineGene 4800 system (BIOER Hangzhou, China, FQD48A). The initial denaturation step was performed at 95 °C for 2 min, followed by 40 cycles of 95 °C for 15 s, 60 °C for 20 s, and elongation. A melting curve analysis (from 60 °C to 95 °C) was conducted to verify the specificity of the PCR products. The tubulin encoding gene tba-1 was used as a reference gene based on microarray data for *C. elegans*. The PCR experiment was repeated at least twice.

## 5. Conclusions

*Lappula patula* extracts demonstrated significant nematocidal and cytotoxic effects in *C. elegans*, including reduced survival, disrupted germline development, and activation of the DNA damage response pathway, leading to apoptosis. These effects, independent of bacterial growth inhibition, highlight the extract’s direct biological impact. This is the first study to use LC-MS to analyze *Lappula patula* extracts, identifying 31 potential anti-tumor compounds. It also provides new insights into its biological effects, such as germline defects, DNA damage responses, and pCHK-1 independent apoptosis in *C. elegans*, which have not been explored previously. These findings suggest that *L. patula* has anti-cancer potential, positioning it as a promising candidate for cancer therapeutics, warranting further research. Future studies, including in vivo validation in mammalian models and further analysis of the most effective anti-tumor compounds, are essential to fully explore its therapeutic potential and mechanisms of action. However, our findings also raise concerns regarding its cytotoxic properties, underscoring the need for a careful evaluation of its safety. Given the broader medicinal potential of the *Lappula* genus, it is crucial to balance its therapeutic benefits with the potential risks through continued research.

## Figures and Tables

**Figure 1 pharmaceuticals-18-00089-f001:**
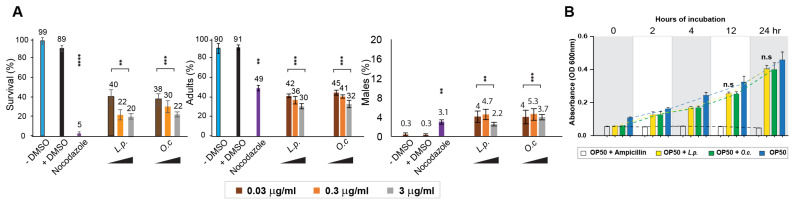
Dose-dependent nematocidal effects of *Lappula patula* extracts on *C. elegans* survival and development, with no impact on OP50 growth. (**A**) *Lappula patula* extracts significantly reduced the survival and development of *C. elegans*. The effects were evaluated by treating worms with different concentrations of *L. patula* extracts (0.03, 0.3, and 3 µg/mL, indicated by brown, orange, and gray colors, respectively) and monitoring their survival, adult formation, and male (HIM) phenotype over a 48 h period. A clear inverse relationship was observed between the dose of the herbal extract and the survival and adult worm percentages, indicating that higher concentrations of *L. patula* extract led to a marked decrease in worm viability and maturation. However, the percentage of males did not exhibit a dose-dependent trend, suggesting that the disruption in sex chromosome segregation may not be directly influenced by the dosage of the extract. Statistical significance was assessed using a two-tailed *t*-test, with ** *p* < 0.01; *** *p* < 0.001; and **** *p* < 0.0001, comparing the control (+DMSO) with the treated samples. (**B**) To evaluate whether the nematocidal effects of *Lappula patula* could be attributed to an inhibition of bacterial growth, we assessed the growth of OP50 in the presence of *L. patula* extract. Over a 24 h incubation period, no significant inhibition of bacterial growth was observed at 0.03 μg/mL of *L. patula* extract.

**Figure 2 pharmaceuticals-18-00089-f002:**
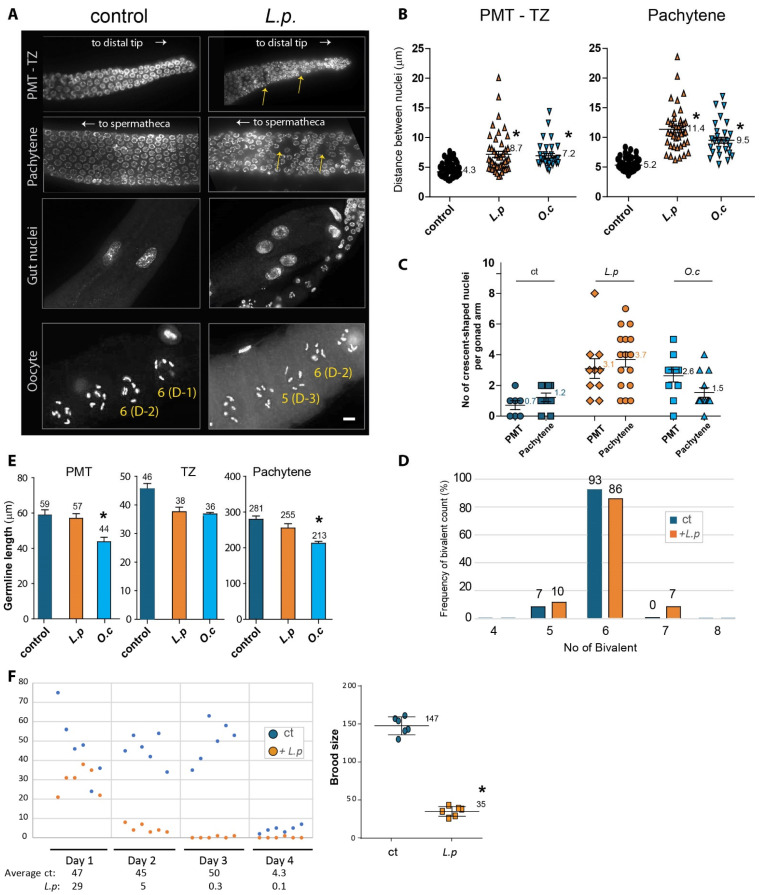
Disruption of germline development caused by *Lappula patula* extracts. (**A**) DAPI-stained nuclei during germline development. Exposure to *Lappula patula* extracts led to an increase in the gaps between nuclei in the PMT and pachytene, as indicated by the arrows. The distances between adjacent nuclei were greater in the worms treated with the herb extract compared with the control (DMSO). Worms exposed to the herbal extract often exhibited fewer DAPI-stained bivalent bodies during diakinesis, suggesting impaired DNA recombination. Scale bar: 2 µm. (**B**) Quantification of the increased nuclear spacing in the premeiotic tip (PMT) and pachytene stages shown in the panel. (**C**) Quantification of crescent-shaped nuclei in the germline. The number of crescent-shaped nuclei per gonad arm is indicated. (**D**) Quantification of DAPI-stained bivalents in the germline. The percentage of bivalent is indicated. (**E**) Quantification of germline size. Germline size, as indicated by the length of the PMT, TZ, and pachytene stages, was measured in worms treated with *L. patula* or *O. cornuta* extracts. (**F**) Brood size of *Lappula patula*-exposed worms. Treatment with *Lappula patula* extracts led to a notable decrease in the number of offspring produced by hermaphrodite worms over a span of four days. Statistical significance was determined using a two-tailed Mann–Whitney test, as indicated by asterisks. All experiments were performed with *C. elegans* hermaphrodites, and the data are presented as the mean ± SEM.

**Figure 3 pharmaceuticals-18-00089-f003:**
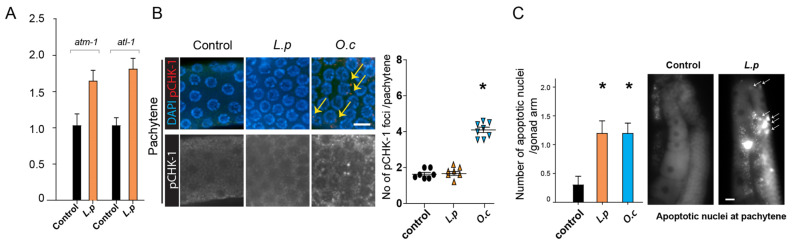
*Lappula patula* extracts activated the DNA damage response, leading to upregulation of *ATM-1* and *ATL-1*, and increased apoptosis but did not induce pCHK-1 foci. (**A**) Exposure to *Lappula patula* extracts significantly enhanced the mRNA expression of *atm-1* and *atl-1*, two essential proteins involved in the DNA damage response pathway, confirming the activation of this cellular defense mechanism. Both *atm-1* and *atl-1* are critical for activating repair pathways such as homologous recombination and cell cycle arrest, and their upregulation in response to extract exposure provides evidence of the activation of these protective mechanisms. This upregulation suggests that the extracts may trigger a cellular reaction to DNA damage, which is necessary to repair the compromised genome. (**B**) No distinct increase in pCHK-1 foci (arrows) was observed in the germline cells of worms treated with *Lappula patula* extracts. Despite the elevation of *atm-1* and *atl-1* levels, no significant increase in pCHK-1 levels was observed, implying that while the DNA damage response was triggered, the downstream signaling associated with checkpoint activation, particularly the phosphorylation of CHK-1, did not occur as expected. The extract from *O.c.* was used as a positive control for these experiments, as it has been previously shown to activate the DNA damage response pathway robustly (*p* = 0.825 in control and *L.p.*, *p* = 0.0021 in control and *O.c.*; scale bar = 2 µm. (**C**) When examined during the pachytene stage, a significant increase in apoptosis (arrow) was detected in the germline cells exposed to *Lappula patula* extracts. Statistical significance was determined by a two-tailed Mann–Whitney test, with asterisks denoting *p* values indicating significant differences between the control and experimental groups. Scale bar = 20 µm.

**Figure 4 pharmaceuticals-18-00089-f004:**
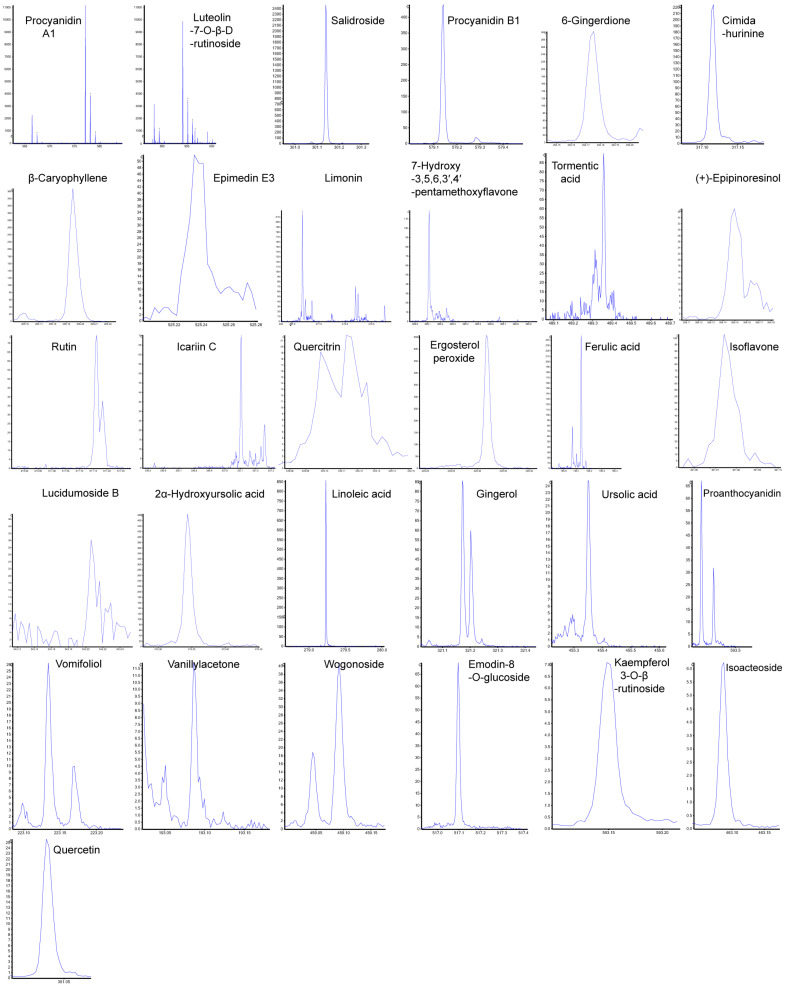
Fragmentation patterns of 31 potential anti-tumor components identified in *Lappula patula* extracts. Out of 112 identified substances, 31 were found to be potential anti-tumor components. The peaks (hollow arrows) shown represent the breakdown patterns of these 31 components, analyzed using PeakView Analyst TF 1.6 software. The *x* axis shows the mass-to-charge ratio (*m*/*z*), and the *y* axis shows the intensity.

**Figure 5 pharmaceuticals-18-00089-f005:**
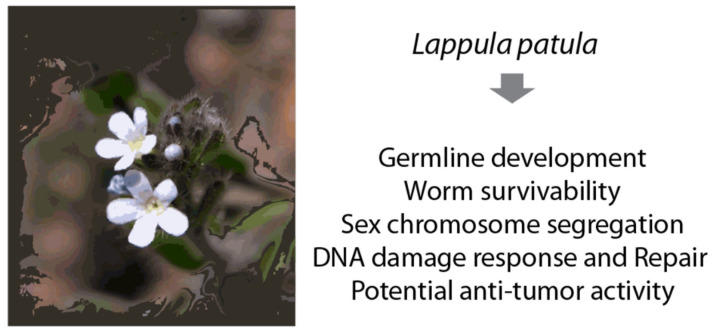
*Lappula patula* extract triggered a DNA damage response, which resulted in disturbances in germline development, enhanced apoptotic cell death, and disruption of the normal meiotic process. These findings suggest that the extract has cytotoxic effects. Furthermore, *Lappula patula* contains at least 31 compounds known for their potential anti-tumor properties. This presence of bioactive compounds underscores the potential of *Lappula patula* as a source of natural anti-cancer agents. Given the growing interest in plant-based treatments for cancer, further investigation into the specific compounds and their mechanisms in *Lappula patula* is essential to fully explore its therapeutic potential.

**Table 1 pharmaceuticals-18-00089-t001:** Summary of identified compounds in *L. patula* based on chemical categories. The full list and information can be found in Appendix A.

Category	Compounds
Flavonoids	Rutin, quercetin-3-o-rutinose, quercetin, isoquercetin, hyperoside, quercitrin, isoquercitrin, luteolin-7-O-β-D-rutinoside, luteolin 7-O-rutinoside, wogonoside, chrysoeriol, isovitexin-2”-O-β-D-glucopyranoside, isoorientin, luteoloside, astragalin, tiliroside
Terpenoids	Citral, 6-gingerdione, vomifoliol, β-caryophyllene, gingerene/α-zingiberene, aromadendrene, ursolic acid, betulinic acid, tormentic acid, 3-ketooleanolic acid, acetyloleanolic acid, 2α-hydroxyursolic acid, limonin
Proanthocyanidins (Tannins)	Procyanidin A1, procyanidin B1, proanthocyanidin
Fatty Acids and Lipids	Linoleic acid, (Z,Z)-9,12-octadecadienoic acid, palmitic acid ethyl ester
Alkaloids	Talatisamine, cimidahurinine, ikarisoside D, ikarisoside E
Stilbenoids	Trans-THSG, cis-THSG
Lignans	(+)-epipinoresinol, isopinoresinol
Steroids and Glycosides	Ergosterol peroxide, oleuropein
Miscellaneous Compounds	Uridine, betaine, dibutyl phthalate, cyclo(Ile-Ala), cyclo(Pro-Ser)

**Table 2 pharmaceuticals-18-00089-t002:** List of 31 anti-tumor compounds identified from *L.p*. extracts. This table highlights compounds with potential anti-tumor properties identified from the *L.p*. extract in heptane using LC-MS. Please see Appendix A for a full description.

No.	Compounds	Chinese Name	Molecular Formula	Mass (Da)
1	**Procyanidin A1**	原花青素A1	C_30_H_24_O_12_	576.12678
2	**Luteolin-7-O-β** **-D-rutinoside**	木樨草素-7-O-β-D-芸香糖苷	C_27_H_30_O_15_	594.15847
3	**Rosavin salidroside**	红景天苷	C_14_H_20_O_7_	300.1209
4	**Procyanidin B1**	原花青素B1	C_30_H_26_O_12_	578.14243
5	**6-gingerdione**	6-姜辣二酮	C_17_H_24_O_4_	292.16746
6	**Cimidahurinine**	北升麻宁	C_14_H_20_O_8_	316.11582
7	**β-caryophyllene**	β-榄香烯	C_15_H_24_	204.1878
8	**Epimedin E3**	淫羊藿黄酮次甙E3	C_26_H_36_O_11_	524.22576
9	**Limonin**	柠檬苦素	C_26_H_30_O_8_	470.19407
10	**7-hydroxy-3,5,6,3′,4′-pentamethoxyflavone**	7-羟基-3,5,6,3′,4′-五甲氧基黄酮	C_20_H_20_O_8_	388.11582
11	**Tormentic acid**	委陵菜酸	C_30_H_48_O_5_	488.35018
12	**(+)-epipinoresinol**	表松脂酚	C_20_H_22_O_6_	358.14164
13	**Rutin**	芦丁	C_27_H_30_O_16_	610.15339
14	**Icariin C**	淫羊藿苷C	C_20_H_16_O_5_	336.09977
15	**Quercitrin**	槲皮甙	C_21_H_20_O_11_	448.10056
16	**Ergosterol peroxide**	过氧麦角甾醇	C_28_H_44_O_3_	428.32905
17	**Ferulic acid**	阿魏酸	C_10_H_10_O_4_	194.05791
18	**Isoflavone**	拟石黄衣醇	C_16_H_12_O_6_	300.06339
19	**Lucidumoside B**	女贞果苷B	C_25_H_34_O_13_	542.19994
20	**2α-hydroxyursolic acid**	2**α**-羟基熊果酸	C_30_H_48_O_4_	472.35526
21	**Linoleic acid**	亚油酸	C_18_H_32_O_2_	280.24023
22	**Gingerol**	姜辣醇	C_19_H_30_O_4_	322.21441
23	**Ursolic acid**	熊果酸	C_30_H_48_O_3_	456.36035
24	**Proanthocyanidin**	原花青素	C_30_H_26_O_13_	594.13734
25	**Vomifoliol**	催吐萝芙木醇	C_13_H_20_O_3_	224.14124
26	**Vanillylacetone**	姜酮	C_11_H_14_O_3_	194.09429
27	**Wogonoside**	汉黄芩苷	C_22_H_20_O_11_	460.10056
28	**Emodin-8-O-(6′-methylpyrrolyl) glucoside**	大黄素-8-O-(6′-甲基丙二酰)吡喃葡萄糖苷	C_24_H_22_O_13_	518.10604
29	**Kaempferol 3-O-β-rutinoside**	莰菲醇-3-O-芸香糖苷	C_27_H_30_O_15_	594.15847
30	**Isoacteoside**	异懈皮苷	C_21_H_20_O_12_	464.09548
31	**Quercetin**	槲皮素	C_15_H_10_O_7_	302.04265

## Data Availability

Data are contained within the article and Appendix A.

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
