# Peer review of "Therapeutic Potential of *Lappula patula* Extracts on Germline Development and DNA Damage Responses in *C. elegans"

_pharmaceuticals, 2025, doi:10.3390/ph18010089_

Round 1

Reviewer 1 Report

Comments and Suggestions for Authors

The manuscript reported effects of Lappula patula extracts in C. elegans. Assays were conducted for its activity in nematocidal, germline disruption, and DNA damage responses. These biologic activities can be observed in many medicinal plant phytochemicals. What are the significant differences between the present findings and the existing results? How would these contribute to the cancer drug development?  Increasing concentrations of L. patula extract led to reduction in worm survival and development, but an alternative approach should be conducted to provide supportive data in the cell check point and the apoptosis. Why did the extract not affect the male phenotype? The toxicity of the extract might pose a high risk in human us. Finally, there are no references cited from the last two years. More update references need to be included.

Comments on the Quality of English Language

Can be improved.

Author Response

Reviewer 1-The manuscript reported effects of Lappula patula extracts in C. elegans. Assays were conducted for its activity in nematocidal, germline disruption, and DNA damage responses. These biologic activities can be observed in many medicinal plant phytochemicals. What are the significant differences between the present findings and the existing results? How would these contribute to the cancer drug development? 

We appreciate the reviewer’s comment. The key difference between the present study and previous research is the depth of analysis. This is the first study to use LC-MS to analyze Lappula patula extracts, identifying 31 potential anti-tumor compounds. It also provides new insights into its biological effects, such as germline defects, DNA damage responses, and pCHK-1 independent apoptosis in C. elegans, which were not previously explored. These findings suggest L. patula has anti-cancer potential, a topic not studied before, and position it as a promising candidate for cancer therapeutics, warranting further research.

We would like to note that we have addressed this in our manuscript. It reads as follows:

Line 424

This research utilized C. elegans to evaluate the nematocidal toxicity of Lappula patula herbal extracts and explore their effects on DNA damage repair and checkpoint responses for the first time (Figure 5). These findings suggest L. patula has anti-cancer potential, a topic not studied before, and position it as a promising candidate for can-cer therapeutics, warranting further research. A screening of 316 herbal extracts re-vealed Lappula patula as a potent inducer of DNA damage checkpoint activation, apoptosis related to DNA damage, HIM phenotypes, impaired meiotic progression, and decreased survival rates. [1, 22].

Line 434

Lappula patula contains at least 31 compounds known for their potential anti-tumor properties. This presence of bioactive compounds underscores the potential of Lappula patula as a source of natural anti-cancer agents. Given the growing interest in plant-based treatments for cancer, further investigation into the specific compounds and their mechanisms in Lappula patula is essential to fully explore its therapeutic potential

Line 519

This is the first study to use LC-MS to analyze Lappula patula extracts, identifying 31 potential anti-tumor compounds. It also provides new insights into its biological effects, such as germline defects, DNA damage responses, and pCHK-1 independent apoptosis in C. elegans, which were not previously explored. These findings suggest L. patula has anti-cancer potential, a topic not studied before, and position it as a promising candidate for cancer therapeutics, warranting further research.

Reviewer 1-Increasing concentrations of L. patula extract led to reduction in worm survival and development, but an alternative approach should be conducted to provide supportive data in the cell check point and the apoptosis. Why did the extract not affect the male phenotype? The toxicity of the extract might pose a high risk in human us.

Thank you for the reviewer's comment. To clarify the reviewer’s suggestion, we have now incorporated the following, and it reads as follows:

 Line 431

The induction of male phenotypes appears to occur only within a certain concentration range, beyond which further increases in dose do not affect the incidence of males. This suggests that male phenotype induction is not a direct consequence of higher doses but may instead result from a saturation effect beyond a specific threshold. In contrast, the dose-dependent larval arrest observed indicates that the herb extract's effects on worm survivability correlate with its concentration. As the dose increases, larval growth is inhibited, leading to reduced survival rates. This supports the idea that the dose of L. patula extract directly impacts larval development and survival, although it does not appear to further influence male incidence at higher doses

Line 527

However, our findings also raise concerns regarding its cytotoxic properties, underscoring the need for a careful evaluation of its safety. Given the broader medicinal potential of the Lappula genus, it is crucial to balance its therapeutic benefits with the potential risks through continued research.

Reviewer 1-Finally, there are no references cited from the last two years. More update references need to be included.

Thank you for the reviewer's comment. In response to the reviewer’s suggestion, we have now incorporated two recent publications that mention L. patula in the manuscript:

Line 42

In some areas, it has been noted as a floristic rarity and regional novelty, with occur-rences in limited or peripheral environments [6].

Line 48

  1. patula differs from its closely related species, L. botschantzevii, in both the size of the corolla and the structure of the inflorescence [8].

Reviewer 2 Report

Comments and Suggestions for Authors

please see the attached file 

Round 2

Reviewer 1 Report

Comments and Suggestions for Authors

The manuscript has been sufficiently amended in accordance to suggestions.

Comments on the Quality of English Language

The manuscript needs English editing.